# H-Mem: Harnessing synaptic plasticity with Hebbian Memory Networks

**Thomas Limbacher and Robert Legenstein**
Institute of Theoretical Computer Science
Graz University of Technology
8010 Graz, Austria
{limbacher,legenstein}@igi.tugraz.at

## Abstract

The ability to base current computations on memories from the past is critical for many cognitive tasks such as story understanding. Hebbian-type synaptic plasticity is believed to underlie the retention of memories over medium and long time scales in the brain. However, it is unclear how such plasticity processes are integrated with computations in cortical networks. Here, we propose Hebbian Memory Networks (H-Mems), a simple neural network model that is built around a core hetero-associative network subject to Hebbian plasticity. We show that the network can be optimized to utilize the Hebbian plasticity processes for its computations. H-Mems can one-shot memorize associations between stimulus pairs and use these associations for decisions later on. Furthermore, they can solve demanding question-answering tasks on synthetic stories. Our study shows that neural network models are able to enrich their computations with memories through simple Hebbian plasticity processes.

## 1 Introduction

Virtually any task faced by humans has a temporal component and therefore demands some form of memory. Consequently, a variety of memory systems and mechanisms have been shown to exist in the brain of humans and other animals [1]. These memory systems operate on a multitude of time scales, from seconds to years. Yet, it is still not well understood how memory is implemented in the brain and how cortical neuronal networks utilize these systems for computation. Most computational models of memory focus on working memory. However, many everyday tasks necessitate a more associative type of memory that acts on a longer time scale. Imagine for example a person reading a book. The person encounters names of people and has to associate these names with specific characteristics and events. As the person continues to read through the book, she has to remember many such associations to build an internal model of the story. This is a veritable computational problem and yet humans are able to solve it seemingly without effort.

Due to its very limited capacity and its short-term nature, working memory cannot satisfy the needs for such tasks. It is widely believed that longer-term storage capabilities are based on Hebbian synaptic plasticity [2]. While it has been shown that Hebbian plasticity can implement auto-associative and hetero-associative memory [3]–[5], it was rarely demonstrated that it can be utilized by neural networks for demanding tasks [6], [7]. There exists abundant evidence that memory is not an automatic process. Rather, cortical networks — in particular networks in prefrontal cortex — are believed to control the storage and retrieval of associations in memory [8]. In this article, we propose a simple network architecture inspired by this idea: the Hebbian Memory Network (H-Mem). The core of H-Mem is a simple hetero-associative network where synapses are subject to Hebbian plasticity. The content to be stored there, and retrieved from there, is defined and prepared by a number of

small sub-networks around that memory. We then train these networks to make optimal use of the Hebbian plasticity in the associative network. We show that this simple architecture is sufficient to solve rather complex tasks that require substantial amounts of memory. We first show that H-Mem can memorize in a single shot associations between stimulus pairs and later use these associations for decisions. Second, we demonstrate that this biologically plausible architecture can solve all of the bAbI tasks [9]. This suite of question answering tasks on synthetic stories was introduced to probe the story-understanding capabilities of deep neural networks. We find that our model outperforms long short-term memory (LSTM, [10]) networks on such tasks and performs comparable to or better than memory-augmented neural networks recently proposed in the machine learning literature [11]–[13].

## 2   Related work

Several models have been proposed for the implementation of working memory capabilities in neural networks. They have been divided into three categories: classical persistent activity, activity silent, and dynamic coding [14]. While models based on persistent activity do not engage any synaptic plasticity mechanism [15], activity-silent models and some dynamic coding models exploit short-term synaptic plasticity or neuronal adaptation [16]–[18]. The use of Hebbian plasticity for working memory has been proposed recently [19].

Associative memory can be implemented in neural networks in two flavors. In a hetero-associative memory, a pattern $k$ is associated with another pattern $v$. The classical model for hetero-associative memory is the Willshaw network [3], which implements this association simply in the connection matrix between two layers of neurons. The Hopfield network is the classical example of an auto-associative network. Here, patterns are associated with themselves in a recurrent neural network [4]. While storage capacity has been thoroughly studied in such models, they have rarely been used for demanding computations.

The idea to utilize rapidly changing ("fast") weights in artificial neural networks for memory was already used in [20], [21] and was adopted recently [6], [7]. In [6], Hebbian plasticity was used to bind input representations to labels for supervised learning. In general, investigations on how to integrate memory into deep learning led to a family of models collectively referred to as memory-augmented neural networks [9], [11], [12], [22]–[25]. The memory module in these models is very much a differentiable version of a digital memory. In contrast, H-Mem is based on an associative memory implemented by Hebbian plasticity. The neural network then does not learn to used the memory module, it rather learns to make use of Hebbian synaptic plasticity. A model that is based on outer-product attention was recently proposed in [26]. Their model utilizes outer products to construct a set of hetero-associative memories representing relationships between items stored in an item memory. Entity Networks [13] have been shown to perform well on the bAbI tasks. Due to heavy use of weight sharing, they are however less attractive as a model for memory utilization in the brain. The bAbI tasks were also tackled using tensor product representations instead of memory control [27]. Metalearned Neural Memory uses a deep neural network as a memory module, in which each layer is updated with a perceptron learning rule using meta-learned targets [28]. Training of networks with synaptic plasticity has been explored in [29], but there the idea was to optimize the parameters of the plasticity rule and not to optimize the control of the plasticity by other neural networks.

## 3   Results

### 3.1   Hebbian Memory Networks

It is widely believed that the brain uses Hebbian synaptic plasticity to store memories over longer time scales. We used the simplest possible implementation of this idea in our model, that is, a hetero-associative memory module which is controlled by simple networks that store and recall information, see Fig. 1 and *Methods* for detailed equations. Consider a set $\mathcal{X}$ of possible inputs to the network. Inputs from $\mathcal{X}$ are presented to the network in a sequential manner. We denote the input at time step $t$ as $x_t \in \mathcal{X}$. Each input $x_t$ is either representing some fact that might be useful to store, or a query to which the network should respond with an output that can be interpreted as an action or an answer. Inputs can be arbitrary objects, we will consider below images and sentences as examples. Inputs are embedded in $d$-dimensional space $\mathbb{R}^d$ using an input encoder, resulting in the

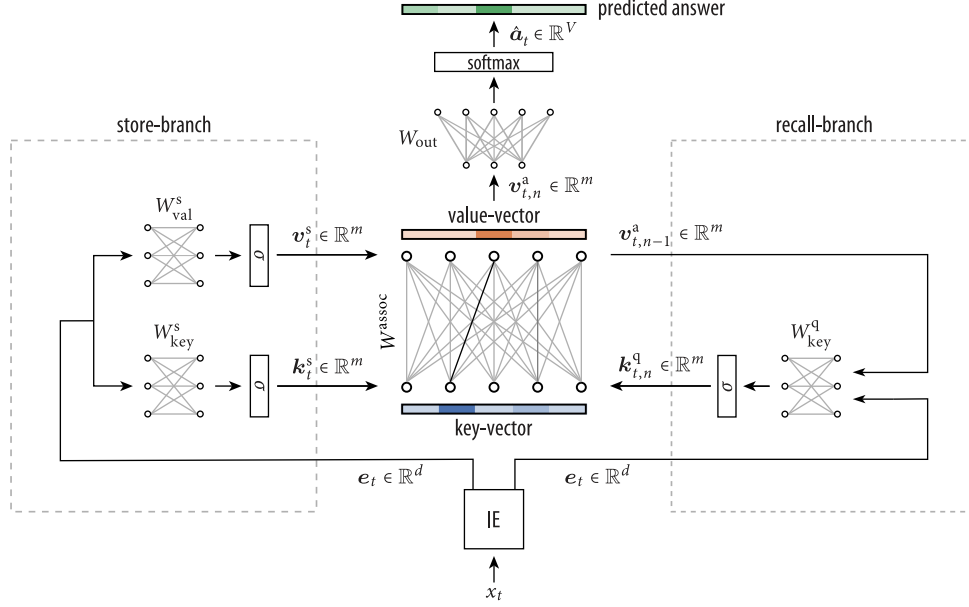

Figure 1: **H-Mem schema.** Inputs $x_t$ are mapped to a continuous space by an input encoder IE (bottom). Depending on whether the input is a fact or a query, either the store-branch (left) or the recall-branch (right) is entered. Store-branch: Two single-layer neural networks with input $e_t$. Network with weight matrix $W_{\text{key}}^{\text{s}}$ outputs a key-vector $k_t^{\text{s}}$ (ReLU nonlinearity denoted as $\sigma$). Network with weight matrix $W_{\text{val}}^{\text{s}}$ outputs a value-vector $v_t^{\text{s}}$. Associations between key-vector and value-vector are stored in the memory matrix $W^{\text{assoc}}$ (middle) through Hebbian plasticity. Recall-branch: A single layer neural network with weight matrix $W_{\text{key}}^{\text{q}}$ gets as input the input embedding $e_t$ and the previously recalled value vector (in the simpler feed-forward model, the latter input is missing). It outputs a key-vector $k_{t,n}^{\text{q}}$ which is used to query the memory matrix for the associated value $v_{t,n}^{\text{a}}$. In the recurrent model, several memory queries can be performed. After $N$ such queries, the final value-vector $v_{t,N}^{\text{a}}$ is passed through the output layer to produce the final answer $\hat{a}_t$.

embedding vector $e_t$. For images, we used a convolutional neural network (CNN) for this embedding, for sentences, we used a simple linear embedding. If the input is a fact, the store-branch is entered (left branch in Fig. 1), which potentially stores some information about $e_t$ in the memory module. In the case of a query, the recall-branch is entered (right branch in Fig. 1), which can recall information based on $e_t$ from the memory module.

In the store-branch, the input-embedding $e_t$ is the input to two single layer neural networks (with ReLU nonlinearities). We call the output activities of this networks the key-vector $k_t^{\text{s}} \in \mathbb{R}^m$ and the value-vector $v_t^{\text{s}} \in \mathbb{R}^m$ respectively. These activity vectors are applied to the memory module, which is a single-layer hetero-associative neural network with matrix $W^{\text{assoc}} \in \mathbb{R}^{m \times m}$. We then use a Hebbian plasticity rule for establishing the association between key-vector $k_t^{\text{s}}$ and value-vector $v_t^{\text{s}}$:

$$\Delta W_{kl}^{\text{assoc}} = \gamma_+ (w^{\text{max}} - W_{kl}^{\text{assoc}}) v_{t,k}^{\text{s}} k_{t,l}^{\text{s}} - \gamma_- W_{kl}^{\text{assoc}} k_{t,l}^{\text{s}}{}^2, \tag{1}$$

where $\gamma_+ > 0$, $\gamma_- > 0$, and $w^{\text{max}}$ are constants. The first term $(w^{\text{max}} - W_{kl}^{\text{assoc}})$ implements a soft upper bound $w^{\text{max}}$ on the weights. The Hebbian term $v_{t,k}^{\text{s}} k_{t,l}^{\text{s}}$ strengthens connections between co-active components in the key- and value-vectors. Finally, the last term generally weakens connections from the currently active key-vector components. Since the Hebbian component strengthens connections to active value-vector components, this emphasizes the current association and de-emphasizes old ones. This update is simlar to Oja's rule [30], but note that the quadratic term acts on the pre-synaptic neuron. In summary, the store branch generates a key- and a value-representation which are associated through Hebbian synaptic plasticity.

We now consider a query input. In this case, the recall-branch is followed. We consider two versions of the model. In the simple feed-forward model, $e_t$ is transformed by a single-layer neural network to a key-vector $k_t^{\text{q}} \in \mathbb{R}^m$ and applied to the memory module. The activation of the key-vector

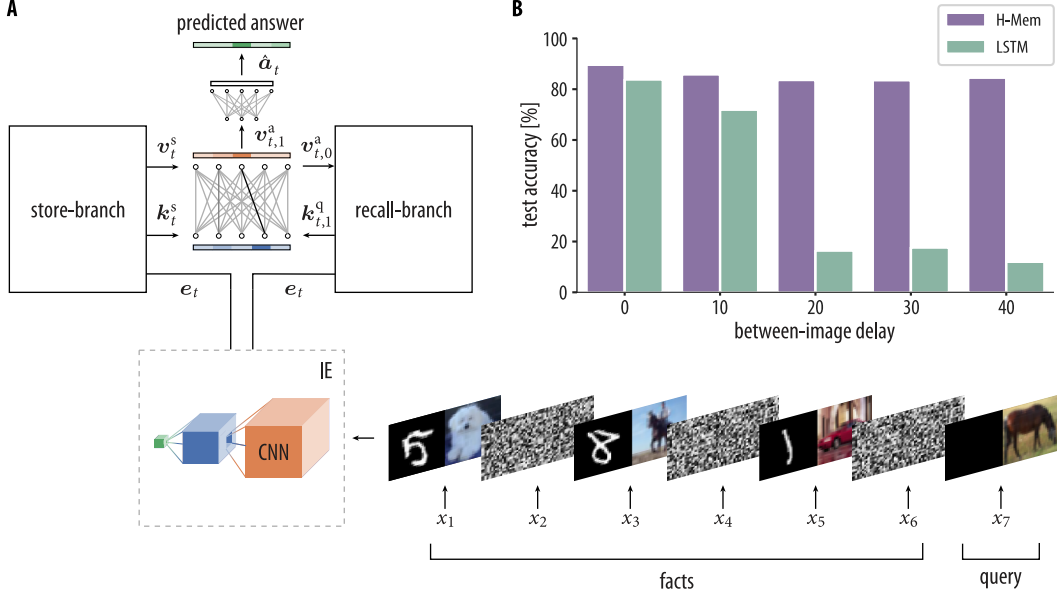

Figure 2: **Single-shot associations with H-Mems. A**) H-mem network with a CNN as input encoder (left) and network input (bottom right). Three image pairs are presented sequentially as facts. During delays between images, random gray-scale images are shown (up to 40 time-steps delay; delay of one time step is shown). Image pairs ($x_1$, $x_3$, and $x_6$) show a handwritten digit (from the MNIST data set) together with an object or an animal (from the CIFAR-10 data set). After all of those images have been presented to the network, it receives a query image ($x_7$). The network is required to output the label of the handwritten digit that appeared together with an image of the same class as the query image. To solve this task the network has to one-shot memorize associations between a handwritten digit and an object or an animal that appear together in an image (and use these associations to give the correct answer). **B**) Comparison of the performance of H-Mem with an LSTM network in this association task. Shown is the test accuracy for various between-image delays. The accuracy of the LSTM model drastically drops at between-image delays of 20 while the H-Mem model's accuracy stays on a constant high level.

neurons in the memory module activates the value neurons, giving rise to the recalled value-vector $\boldsymbol{v}_t^{\mathrm{a}} = W^{\mathrm{assoc}} \boldsymbol{k}_t^{\mathrm{q}}$. The model output $\hat{\boldsymbol{a}}_t$ is then given by another neural layer followed by a softmax. In the second version of the model, we consider a recurrent recall-branch. In this case, $N$ recalls can be initiated before the output is given. More precisely, at recall-step $n$, the computation of the key-vector $\boldsymbol{k}_{t,n}^{\mathrm{q}}$ is based not only on the input embedding $\boldsymbol{e}_t$, but also on the value-vector of the previous recall $\boldsymbol{v}_{t,n-1}^{\mathrm{a}}$ (see the "recall-branch" in Fig. 1; for the first recall, the zero-vector is used for the value-vector). After $N$ such recalls, we compute the output $\hat{\boldsymbol{a}}_t$ of the network based on $\boldsymbol{v}_{t,N}^{\mathrm{a}}$.

In order to test whether the control networks can make use of the Hebbian plasticity in the memory module, we trained the network end-to-end with gradient descent on a variety of tasks (see below), where the correct response to queries was provided as the target network output. The weight matrices of the network $W_{\mathrm{key}}^{\mathrm{s}}$, $W_{\mathrm{val}}^{\mathrm{s}}$, $W_{\mathrm{key}}^{\mathrm{q}}$, and $W_{\mathrm{out}}$ (see Fig. 1) were optimized. Note that the association matrix $W^{\mathrm{assoc}}$ of the memory module was not optimized. This matrix is a dynamic variable just like neuron activations and is updated according to Eq. (1) after the key- and value-vector have been computed in the store-branch (see *Methods*).

### 3.2 Flexible associations through Hebbian plasticity

To test the ability of the model to one-shot memorize associations and to use these associations later when needed, we conducted experiments on a task that requires to form associations between entities that appear together in a sequence of images.

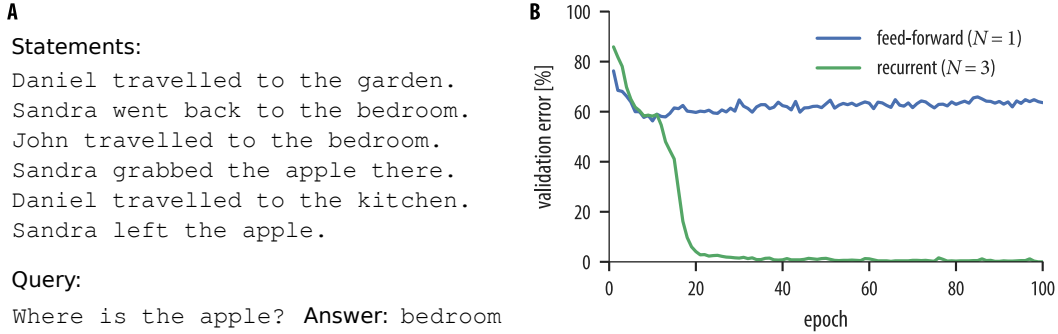

**A**

Statements:

```
Daniel travelled to the garden.
Sandra went back to the bedroom.
John travelled to the bedroom.
Sandra grabbed the apple there.
Daniel travelled to the kitchen.
Sandra left the apple.
```

Query:

`Where is the apple?` **Answer:** `bedroom`

Figure 3: **Sample story from task 2 of the bAbI data set and the evolution of the validation error of H-Mem for this task. A**) Example story from task 2 of the bAbI data set. It contains a set of statements, a question, and the corresponding answer. One difficulty of the task is that several statements and their relative temporal order have to be considered to answer the question. **B**) Evolution of the validation error over 100 epochs of the feed-forward (blue) and recurrent (green) H-Mem model for task 2 of the bAbI data set. Since multiple supporting statements have to be chained to answer the questions in this task, the feed-forward model fails to solve it. The recurrent model, performing $N = 3$ memory queries, easily solves this task (results are shown for our models with LE in the 10k training examples setting). Further examples of bAbI tasks along with the validation error of our H-Mem models on these tasks are shown in Fig. S1 of the Supplementary.

In each instance of this task, we randomly chose three classes out of the 10 classes of the CIFAR-10 data set and randomly associated a unique digit (between 0 and 9) to each. For example, we associated the digit 5 with the class "dogs", the digit 8 with the class "horses", and the digit 1 with the class "cars" (see Fig. 2A, bottom). We then generated images that showed a randomly drawn instance of the CIFAR-10 class on the right and a randomly drawn example of the associated digit from the MNIST data set on the left. After these images have been presented to the network, it received an additional query image. The query image showed another randomly drawn sample from one of the previously shown CIFAR-10 classes, but no handwritten digit. The network was required to output the digit that was temporarily associated with the query image class. Note that the classes, associated digits, and specific class images were freshly drawn for each example. Hence, the network had to store the association in its memory during inference. To increase the difficulty of the task, we added a delay of up to 40 time steps after each image, during which the network received (up to 40) random gray-scale images (see Fig. 2A, bottom).

We trained a feed-forward H-Mem network on this task (i.e., the number of memory queries $N$ was set to 1). The network had a hetero-associative memory with a square association matrix of order $m = 200$. It was trained for 100 epochs with a batch size of 32. We used a CNN as input encoder in this task (Fig. 2A). The input encoder summarizes each input image with a dense feature vector of size $d = 128$ (see Supplementary). The error was computed when the network produced its prediction after the query and gradients were propagated through all time steps of the computation.

In Fig 2B we compare our model to a standard LSTM network, using an increasing delay between presented images. The LSTM network consisted of one layer of 200 LSTM units with the same CNN architecture as input encoder. For small between-image delays, the performance of the LSTM network is comparable to H-Mem's performance (although H-Mem is slightly better). While the accuracy of the LSTM network drastically drops at some point, H-Mem's performance is largely insensitive to the considered between-image delays.

### 3.3 Question answering through Hebbian plasticity

Story understanding necessitates rapid and flexible memory on long time scales. A system understands a story, if it can correctly answer questions after it has been exposed to it. Hence, story understanding can be probed with question answering (QA) tasks on stories. To test how well H-Mem can utilize Hebbian plasticity for story understanding, we evaluated the model on the bAbI data set [9]. The data set contains 20 different types of synthetic QA tasks, designed to test a variety of reasoning abilities on stories. Each of these QA tasks consist of a sequence of statements followed by a question whose

answer is typically a single word (in a few task, answers are multiple words). We provided the answer to the model as supervision during training, and it had to predict it at test time on a separate test set. Note that statements that provide the answer to a question are given among other statements which are irrelevant for that question (see Figure 3A for an example). Since the question is given as the last sentence and we did not indicate to the model which of the statements are relevant, it had to store many facts and infer from the query which facts to combine from memory in order to answer the question. The performance of the model was measured using the average error on the test data set over all tasks and the number of failed tasks (the model had failed to solve a task if the test error was above $5\,\%$ for that task).

Each instance of a task consists of a sequence of $T$ sentences $\langle x_1, \ldots, x_T \rangle$ (with $T \leq 321$), where the last sentence is a question, and an answer $a$. We represent each word $j$ in a given sentence $x_t$ by a one-hot vector $\boldsymbol{w}_{t,j}$ of length $V$ (where $V$ is the vocabulary size). In contrast to previous work ([11], [13]), we did not limit the number of sentences in a story.

We evaluated three different input embeddings (IE in Fig. 1) for sentences. The first one is the standard bag-of-words (BoW) representation (see Supplementary). As pointed out in [11], this representation has the drawback that it cannot capture the order of the words in the sentence, which is important for some tasks. We therefore used a position encoding (PE) representation that encodes the position of the words within a sentence (see [11] and Supplementary). As a third option, we found it helpful to let the model choose for itself which type of sentence encoding to use. We therefore also evaluated a learned encoding (LE, see Supplementary and [13]). In order to enable our models to capture the temporal context of sentences, we also used a temporal encoding for sentences as introduced in [11] (see Supplementary).

We trained an H-Mem network with an embedding dimension $d$ of $80$ and a hetero-associative memory with a square association matrix of order $m = 100$. We used the recurrent model with the number of memory queries $N$ set to 3. The networks were trained for 100 epochs on $10\,000$ examples per task with a batch size of 128 (see Supplementary for details to the model and the training setup). Similar to previous work [11], [13], we performed three independent runs with different random initializations and report the results of the model with the highest validation accuracy in these runs.

In Table 1 we compare our model to various other alternative models in the literature: an LSTM network [9], the Dynamic Neural Turing Machine (D-NTM) [12], the End-to-End Memory Network (MemN2N) [11], and the Entity Network (EntNet) [13]. We compare the performance of these models in terms of their mean error, error on individual tasks, and the number of failed tasks (results of the alternative models were taken from the respective papers). When considering H-Mem with various input encodings, we found that the position encoding representation gives a clear improvement over bag-of-words, as demonstrated on tasks 4, 5, 15 and 18, where word ordering is particularly important. The learned encoding (LE) is clearly the best input embedding for the H-Mem model. With this input encoding, H-Mem solves 16 of the 20 tasks. H-Mem typically struggles with tasks that demand additional reasoning on memorized information (tasks 16, 17, and 19). We will show below that this can be resolved with a memory-dependent memorization strategy. H-Mem with any type of input encoding outperforms the LSTM network and the D-NTM (the D-NTM is slightly better in the mean error for BoW encoding). With the learned encoding, it is nearly as good as the MemN2N model (with just one solved task difference), despite the fact that it uses a very simple biologically plausible associative memory with Hebbian plasticity. To test how well the model performs on fewer training examples, we also evaluated its performance on a version of the data set with 1000 training examples (as done in [9], [11], [13]). While Entitiy Networks solve all tasks when trained with 10k examples, H-Mem with PE and LE outperforms the EntNet model in the 1k training example setting. This indicates that the simpler H-Mem model is better suited for smaller data sets (see Table S1 in the Supplementary for test error rates on all 20 bAbI tasks for 1k training examples).

In Fig. 3B we compare the performance of the feed-forward model to that of the recurrent version on bAbI task 2. The recurrent model outperforms the feed-forward model in this task. The feed-forward model fails to solve this task since it can perform only one memory query, while this task requires chaining of multiple statements, which can be done only by subsequent memory queries. In general, we observed a variety of tasks that could be solved by our recurrent model but not by the feed-forward model (see Table S2 in the Supplementary for detailed results).

We hypothesized that the model learns (a) to extract the relevant key-value pairs from facts, stores that in memory, and (b) to extract essential keys from queries in order to retrieve the informative

Table 1: Test error rates (in %) on the 20 bAbI QA tasks for models using 10k training examples (botom: mean test errors for 1k training examples). Keys: BoW = bag-of-words representation; PE = position encoding representation LE = learned encoding. Results of the MemN2N model (for 10k training examples) are given for the larger model reported in the Appendix of [11].

| | Baseline | | | | H-Mem | | |
|---|---|---|---|---|---|---|---|
| Task | LSTM | D-NTM | MemN2N | EntNet | BoW | PE | LE |
| 1: Single Supporting Fact | 0.0 | 4.4 | 0.0 | 0.0 | 0.0 | 0.0 | 0.0 |
| 2: Two Supporting Facts | 81.9 | 27.5 | 0.3 | 0.1 | 0.2 | 0.0 | 0.2 |
| 3: Three Supporting Facts | 83.1 | 71.3 | 2.1 | 4.1 | 30.5 | 24.9 | 26.9 |
| 4: Two Arg. Relations | 0.2 | 0.0 | 0.0 | 0.0 | 37.8 | 0.0 | 0.0 |
| 5: Three Arg. Relations | 1.2 | 1.7 | 0.8 | 0.3 | 11.6 | 1.8 | 1.3 |
| 6: Yes/No Questions | 51.8 | 1.5 | 0.1 | 0.2 | 1.2 | 1.5 | 1.2 |
| 7: Counting | 24.9 | 6.0 | 2.0 | 0.0 | 0.5 | 6.8 | 0.8 |
| 8: Lists/Sets | 34.1 | 1.7 | 0.9 | 0.5 | 0.7 | 0.8 | 0.5 |
| 9: Simple Negation | 20.2 | 0.6 | 0.3 | 0.1 | 2.9 | 6.6 | 3.3 |
| 10: Indefinite Knowledge | 30.1 | 19.8 | 0.0 | 0.6 | 1.4 | 1.5 | 1.5 |
| 11: Basic Coreference | 10.3 | 0.0 | 0.0 | 0.3 | 0.0 | 0.0 | 0.0 |
| 12: Conjunction | 23.4 | 6.3 | 0.0 | 0.0 | 0.0 | 0.0 | 0.0 |
| 13: Compound Coref. | 6.1 | 7.5 | 0.0 | 1.3 | 0.0 | 0.0 | 0.0 |
| 14: Time Reasoning | 81.0 | 17.5 | 0.2 | 0.0 | 0.0 | 0.3 | 1.1 |
| 15: Basic Deduction | 78.7 | 0.0 | 0.0 | 0.0 | 10.6 | 0.0 | 0.0 |
| 16: Basic Induction | 51.9 | 49.7 | 51.8 | 0.2 | 53.6 | 54.3 | 54.8 |
| 17: Positional Reasoning | 50.1 | 1.3 | 18.6 | 0.5 | 38.7 | 41.1 | 28.7 |
| 18: Size Reasoning | 6.8 | 0.2 | 5.3 | 0.3 | 44.3 | 6.8 | 1.9 |
| 19: Path Finding | 90.3 | 39.5 | 2.3 | 2.3 | 74.8 | 70.0 | 77.1 |
| 20: Agent's Motivations | 2.1 | 0.0 | 0.0 | 0.0 | 0.0 | 0.0 | 0.0 |
| Mean error | 36.4 | 12.8 | 4.2 | 0.5 | 15.4 | 10.8 | 10.0 |
| Failed tasks (err. $> 5\,\%$) | 16 | 9 | 3 | 0 | 8 | 7 | 4 |
| On 1k training data | | | | | | | |
| Mean error | 51.3 | - | 13.9 | 29.6 | 31.0 | 26.4 | 25.9 |
| Failed tasks (err. $> 5\,\%$) | 20 | - | 11 | 15 | 15 | 13 | 13 |

values. We performed an analysis to test this idea by computing the cosine similarity of the recall key (resp. the recalled value) to keys (and values) of previous storing operations in bAbI task 1. The similarity of keys was $0.996 \pm 0.004$ for sentences with the same person as the person in the query ($0.020 \pm 0.028$ otherwise). For values, the similarity was $0.981 \pm 0.026$ for sentences with the same place as the answer place ($0.323 \pm 0.119$ otherwise). This indicates that the model learned to associate persons to places (see Supplementary for details).

To summarize, we found that Hebbian synaptic plasticity is sufficient to solve rather demanding question-answering tasks on stories. H-Mem can compete with previous deep-learning approaches on theses tasks that used a more computer-like digital memory module. We further found that several memory recalls are necessary to solve most tasks in this domain.

## 3.4 Memory-dependent memorization

In the model considered above, since the computation of key- and value-vectors in the store-branch depends solely on the current input, new key-value pairs are stored in memory without taking the memory content into account. Here, we present an extension to the H-Mem model where the computation of the value-vector during storage is dependent on the memory content. More specifically, in the store-branch we compute — just like before — a key-vector $\boldsymbol{k}_t^{\mathrm{s}}$ and a value-vector $\hat{\boldsymbol{v}}_t^{\mathrm{s}}$ from the input embedding $\boldsymbol{e}_t$. Then, however, we read from memory once, using the key-vector $\boldsymbol{k}_t^{\mathrm{s}}$ to obtain the associated value $\tilde{\boldsymbol{v}}_t^{\mathrm{s}}$. Vectors $\hat{\boldsymbol{v}}_t^{\mathrm{s}}$ and $\tilde{\boldsymbol{v}}_t^{\mathrm{s}}$ are then concatenated and passed through a linear layer to produce a vector $\boldsymbol{v}_t^{\mathrm{s}}$. We then use Eq. (1) for establishing the association between key-vector $\boldsymbol{k}_t^{\mathrm{s}}$ and the new value-vector $\boldsymbol{v}_t^{\mathrm{s}}$.

Table 2: Mean test error rates (in %) on the 20 bAbI QA tasks for models using 10k training examples (top) and 1k training examples (bottom). H-Mem results are given for the extended model with memory-dependent memorization and learned encoding (LE). Results of the MemN2N model (for 10k training examples) are given for the larger model reported in the Appendix of [11].

|  | LSTM | D-NTM | MemN2N | EntNet | H-Mem |
|---|---|---|---|---|---|
| Mean error | 36.4 | 12.8 | 4.2 | 0.5 | 0.6 |
| Failed tasks (err. $> 5\,\%$) | 16 | 9 | 3 | 0 | 0 |
| On 1k training data |  |  |  |  |  |
| Mean error | 51.3 | - | 13.9 | 29.6 | 22.9 |
| Failed tasks (err. $> 5\,\%$) | 20 | - | 11 | 15 | 12 |

We evaluated this extended model on the bAbI tasks. The networks were trained for 250 epochs on 10 000 examples per task with a batch size of 128 and learned encoding (see Supplementary for more details to the training setup). As above, we performed three independent runs with different random initializations and report the results of the model with the highest validation accuracy in these runs (the results are summarized in Table 2, see Table S3 in the Supplementary for results on the individual tasks).

H-Mem solves all the bAbI tasks with a mean error over all 20 tasks of $0.6\,\%$, outperforming the results of MemN2N [11]. It also outperforms the LSTM network of [9] by $35.8\,\%$ and the D-NTM [12] by $12.2\,\%$ in terms of mean error. H-Mem performs almost as good as the EntNet [13] model (with just $0.1\,\%$ difference in the mean error). We also evaluated our extended model in the 1k training example setting where it outperforms the EntNet model in both, the number of solved tasks and the mean error (see Table 2; for results on individual tasks see Table S3 in the Supplementary).

## 4    Methods

The model was implemented in TensorFlow (the code it available at https://github.com/IGITUGraz/H-Mem). Our model takes a sequence of inputs $\langle x_1, \ldots, x_T \rangle$. Each input $x_t$ at time step $t$ can either be a fact or a query to which the network should respond with an answer $\hat{a}_t$.

**Input encoder**    Let $\langle x_1, \ldots, x_t \rangle$ be the given input sequence. Each $x_t$ is converted into a vector $\boldsymbol{e}_t \in \mathbb{R}^d$ by mapping it to a continuous embedding space using an input encoder (IE). We do not restrict the type of the input encoder. It has to be chosen for the task at hand. Typical choices include a CNN, in the case of inputs being images, or a simple linear embedding for sentences.

Depending on whether the input $x_t$ represents some fact or a query, the input in the embedding space, that is $\boldsymbol{e}_t$, enters either the store-branch or the recall-branch. The other branch is then completely inactive in that time step $t$. We start by describing the store-branch.

**Store-branch**    In the store-branch, we compute a vector $\boldsymbol{k}_t^{\text{s}}$ by passing the input-embedding $\boldsymbol{e}_t$ through a weight matrix $W_{\text{key}}^{\text{s}}$ of size $m \times d$ followed by a ReLU nonlinearity:

$$\boldsymbol{k}_t^{\text{s}} = \text{ReLU}(W_{\text{key}}^{\text{s}} \boldsymbol{e}_t), \tag{2}$$

where $\text{ReLU}(\boldsymbol{z}) = (\text{ReLU}_1(\boldsymbol{z}), \ldots, \text{ReLU}_m(\boldsymbol{z}))^{\intercal}$ and $\text{ReLU}_i(\boldsymbol{z}) = \max(0, z_i)$. We call $\boldsymbol{k}_t^{\text{s}}$ the key-vector. Similarly, we compute a value-vector $\boldsymbol{v}_t^{\text{s}}$ by using another matrix $W_{\text{val}}^{\text{s}}$ with the same size as $W_{\text{key}}^{\text{s}}$:

$$\boldsymbol{v}_t^{\text{s}} = \text{ReLU}(W_{\text{val}}^{\text{s}} \boldsymbol{e}_t). \tag{3}$$

We then use a Hebbian plasticity rule for establishing the association between key-vector $\boldsymbol{k}_t^{\text{s}}$ and value-vector $\boldsymbol{v}_t^{\text{s}}$. The associative memory module is represented by a matrix $W^{\text{assoc},t}$ of size $m \times m$. Note that the association weight matrix actually depends on $t$ (this was suppressed in the notation in Results for simplicity). Here, we indicate the time-step as a superscript of the matrix. Weight changes are given by:

$$\Delta W_{kl}^{\text{assoc},t} = \gamma_+ (w^{\text{max}} - W_{kl}^{\text{assoc},t}) v_{t,k}^{\text{s}} k_{t,l}^{\text{s}} - \gamma_- W_{kl}^{\text{assoc},t} {k_{t,l}^{\text{s}}}^2, \tag{4}$$

where $\gamma_+ > 0$, $\gamma_- > 0$, and $w^{\mathrm{max}}$ are constants. The association weight matrix is then updated according to $W^{\mathrm{assoc},t+1} = W^{\mathrm{assoc},t} + \Delta W^{\mathrm{assoc},t}$.

**Recall-branch**    We now consider a query input. Here, the recall-branch is followed and the vector $\boldsymbol{e}_t$ representing the query in the embedding space and a vector $\boldsymbol{v}^{\mathrm{a}}_{t,n-1}$ are concatenated and passed through a matrix $W^{\mathrm{q}}_{\mathrm{key}}$ of size $m \times d + m$ followed by a ReLU nonlinearity to compute the key-vector $\boldsymbol{k}^{\mathrm{q}}_{t,n}$ (we use $t, n$ to index the key- and value-vectors at the $n$th memory query in time step $t$):

$$\boldsymbol{k}^{\mathrm{q}}_{t,n} = \mathrm{ReLU}(W^{\mathrm{q}}_{\mathrm{key}}(\boldsymbol{e}_t^{\mathsf{T}}, \boldsymbol{v}^{\mathrm{a}}_{t,n-1}{}^{\mathsf{T}})^{\mathsf{T}}), \tag{5}$$

The initial value-vector, that is $\boldsymbol{v}^{\mathrm{a}}_{t,0}$, is set to the zero-vector. The key-vector $\boldsymbol{k}^{\mathrm{q}}_{t,n}$ at time $t$ is then used to extract the associated value-vector from memory by taking the matrix-vector product of $W^{\mathrm{assoc},t}$ and the key-vector $\boldsymbol{k}^{\mathrm{q}}_{t,n}$: $\boldsymbol{v}^{\mathrm{a}}_{t,n} = W^{\mathrm{assoc},t} \boldsymbol{k}^{\mathrm{q}}_{t,n}$.

**Generating the final prediction**    The queried value after $N$ queries represented by the vector $\boldsymbol{v}^{\mathrm{a}}_{t,N}$ is then passed through a final weight matrix $W_{\mathrm{out}}$ of size $V \times m$ and a softmax to produce the predicted answer:

$$\hat{\boldsymbol{a}}_t = \mathrm{softmax}(W_{\mathrm{out}} \boldsymbol{v}^{\mathrm{a}}_{t,N}), \tag{6}$$

where $\mathrm{softmax}(\boldsymbol{z}) = (\mathrm{softmax}_1(\boldsymbol{z}), \ldots, \mathrm{softmax}_m(\boldsymbol{z}))^{\mathsf{T}}$ and $\mathrm{softmax}_i(\boldsymbol{z}) = e^{z_i} / \sum_j e^{z_j}$. The weights $W^{\mathrm{s}}_{\mathrm{key}}$, $W^{\mathrm{s}}_{\mathrm{val}}$, $W^{\mathrm{q}}_{\mathrm{key}}$, and $W_{\mathrm{out}}$ are learned during training by minimizing the cross-entropy loss between $\hat{\boldsymbol{a}}$ and the true answer $\boldsymbol{a}$ using the Adam optimizer [31]. The associative memory matrix $W^{\mathrm{assoc}}$ is initialized for each input sequence $\langle x_t \rangle$ with all its values set to zeros. A schema of the model is shown in Fig. 1A.

## 5    Discussion

In this article, we have asked whether neural networks can harness Hebbian synaptic plasticity for computations that demand memories on longer time scales. We found that a rather simple model is sufficient to tackle an association task as well as question-answering tasks on synthetic stories. This shows that Hebbian plasticity is a mechanism that can enrich neural network computations with longer memories. In particular, we found that H-Mem outperforms LSTM networks in many such tasks, indicating that the more long-term nature of synaptic plasticity is superior to the activity-based memory of LSTM networks in the considered situations.

We have optimized our networks with gradient descent, using a variant of the backpropagation algorithm. This algorithm is not biologically plausible. Hence, our study does not allow us to directly conclude anything about how such tasks could be learned in biological neural systems. We hypothesize that evolutionary processes may have evolved circuitry for some very important tasks and that further plasticity may be used to fine-tune the circuitry. Alternatively, recently proposed biologically plausible variants to backpropagation may be interesting in this context [32]–[34]. It remains to be tested however whether these variants are powerful enough to train H-Mem networks.

While we only considered simple rate-based neuron models, we do not see any fundamental reason why these ideas could not be implemented in spiking neural networks. This would also be an interesting step in order to implement H-Mem on energy-efficient spike-based neuromorphic hardware. Once a network its trained, it can make use of completely local synaptic plasticity, a feature that is implemented in currently available neuromorphic systems [35], [36].

We did deliberately not relate the model to brain anatomy, as the organization of higher-level cognitive functions is still very much unknown. However, Hebbian plasticity is well-supported by many experimental findings. In particular the Hippocampus might play a pivotal role for implementing a memory module as in our model [37].

Memory-augmented neural networks have shown that some type of memory can strongly enrich the computational capabilities of neural networks. Previously, this was achieved with rather unbiological types of memory components [11], [22] or with heavy weight sharing [13]. In this article, we have demonstrated that such enrichment is also possible with Hebbian plasticity, one of the most fundamental plasticity principles in biological neuronal systems. In conclusion, our study provides a first link between research on memory-augmented neural networks and biologically plausible models of cognition.

## Broader Impact

This work does not present any foreseeable societal consequence.

## Acknowledgments and Disclosure of Funding

This project has received funding from the Austrian Science Fund, Project #I3251-N33, and Project #I4670-N (SMALL; CHIST-ERA ERA-Net), and the European Union's Horizon 2020 Research and Innovation Programme under Grant Agreement #785907 (HBP SGA2). We thank Wolfgang Maass and Arjun Rao for helpful discussions and ideas. The authors declare that the research was conducted in the absence of any commercial or financial relationships that could be construed as a potential conflict of interest.

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
