[Supplementary Material]

# Supplementary information for: *H-Mem: Harnessing synaptic plasticity with Hebbian Memory Networks*

**Thomas Limbacher and Robert Legenstein**
Institute of Theoretical Computer Science
Graz University of Technology
8010 Graz, Austria
{limbacher,legenstein}@igi.tugraz.at

## 1   Model and training details

Here we give details to our models, and to the encoding and representation of images and questions used in our models.

### 1.1   Details to: Flexible associations through Hebbian plasticity

**Model details**   We used a CNN as input encoder in this task. The CNN consisted of 11 weight layers with the following structure: $5 \times$ (Conv2D $\rightarrow$ BatchNorm $\rightarrow$ Conv2D $\rightarrow$ BatchNorm $\rightarrow$ MaxPooling $\rightarrow$ Dropout) $\rightarrow$ Fully connected $\rightarrow$ BatchNorm $\rightarrow$ Dropout, with 32, 64, 128, 256, and 512 filters, respectively. Each with a kernel size of $3 \times 3$ (stride = 1, padding="same") and ELU nonlinearity [1]. We used a $2 \times 2$ pool size in the max pooling layers. The dropout rate was set to 0.1, 0.1, 0.2, 0.3, 0.3, and 0.3, respectively. The last fully connected layer was of size 128 followed by a ReLU nonlinearity (BatchNorm denotes a batch normalization layer [2]).

**Training details**   We used the MNIST and the CIFAR-10 data set in this task (we kept the default train-test split of these data sets; data sets from the TensorFlow data set API). Training examples were generated as described in the main text. We trained on 11 250 examples and tested on 2230 examples. Here, an example is one full sequence of image pairs (including random images) and one query image. The optimal hyper-parameters were selected through grid search on a held-out validation set which was $10\%$ of the training set. The model was trained with Adam [3] using a learning rate of $\mu = 0.001$, that was decayed exponentially (starting at epoch 50) with a decay rate of 0.1. The weights were initialized using the He uniform variance scaling initializer [4]. We applied $L_2$ regularization to the weights. The $L_2$-norm of these weights was scaled by 0.001 before adding it to the loss. The hetero-associative memory module was represented by a square matrix of order $m = 200$ and was initialized with all its elements set to zero. Plasticity coefficients were set to $\gamma_+ = 0.01$ and $\gamma_- = 0.01$, and $w^{\max}$ was set to 1. The networks were trained for 100 epochs with a batch size of 32 (one epoch consists of all samples of the training set). Gradients with an $L_2$-norm larger than 10.0 were normalized to have norm 10.0. We performed two independent runs with different random initializations and report the results of the model with the highest validation accuracy in these runs.

### 1.2   Details to: Question answering through Hebbian plasticity

**Model details**   We evaluated three different representations for the sentences. The first one is the standard bag-of-words (BoW) representation. It embeds each word $\boldsymbol{w}_{t,j}$ of a sentence $x_t = \{\boldsymbol{w}_{t,1}, \boldsymbol{w}_{t,2}, \ldots, \boldsymbol{w}_{t,J}\}$ and sums the resulting vectors: $\boldsymbol{e}_t = \sum_j A\boldsymbol{w}_{t,j}$. Here, $A$ is the embedding matrix. As [5] pointed out, this representation has the drawback that it can not capture the order of the words in the sentence, which is important for some tasks. We therefore used a representation

that encodes the position of the words within a sentence (as proposed in [5]). The authors call this type of representation position encoding (PE), which takes the form: $\boldsymbol{e}_t = \sum_j \boldsymbol{l}_j \circ A\boldsymbol{w}_{t,j}$, where $\circ$ is the Hadamard product. The column vector $\boldsymbol{l}_j$ with one-based indexing has the structure $l_{kj} = (1 - j/J) - (k/d)(1 - 2j/J)$, where $J$ is the number of words in the sentence and $d$ the embedding size. We found it helpful to let the model choose for itself which type of sentence encoding to use. As proposed in [6], we therefore used a learned encoding (LE) given by $\boldsymbol{e}_t = \sum_j \boldsymbol{f}_j \circ A\boldsymbol{w}_{t,j}$. The vectors $\boldsymbol{f}_j$ were constant across time steps and were trained jointly with the other parameters of our model. By using this type of encoding the model can adapt the sentence representation to best suit the task at hand. It can either choose a BoW representation (by setting all elements in $\boldsymbol{f}_j$ to one), a position encoding, or any encoding beneficial to the task.

In order to enable our models to capture the temporal context of a task, we used a temporal encoding for sentences as introduced in [5]. This encoding uses a special matrix $T_A$ that encodes temporal information. The modified sentence representation is then given by $\boldsymbol{e}_t = \sum_j A\boldsymbol{w}_{t,j} + \text{row}_t(T_A)$ (BoW), $\boldsymbol{e}_t = \sum_j \boldsymbol{l}_j \circ A\boldsymbol{w}_{t,j} + \text{row}_t(T_A)$ (PE), and $\boldsymbol{e}_t = \sum_j \boldsymbol{f}_j \circ A\boldsymbol{w}_{t,j} + \text{row}_t(T_A)$ (LE), where $\text{row}_t(T_A)$ is the $t$th row of the matrix $T_A$. Note that $T_A$ was learned during training and that sentences are indexed in reverse order, so that $x_1$ is the last sentence of a story.

Answers to questions in the bAbI QA tasks are typically a single word. In a few tasks, answers are a set of words (e.g., task 8: Lists/Sets). In this case, we considered each answer as one word in the vocabulary (i.e., there was one output class for each word pair that could be a target output).

We found it helpful to apply a batch normalization layer at the output of the input encoder of our model.

**Training details**  The optimal hyper-parameters were selected through grid search on a held-out validation set which was $10\,\%$ of the bAbI training set. We used version 1.2 of the data set (we kept the default train-test split of the data set). The model was trained with Adam [3] using a learning rate of $\mu = 0.003$, that was reduced by $15\,\%$ every 20 epochs. The weights and the embedding matrices were initialized using the He uniform variance scaling initializer [4]. We found it helpful to apply $L_2$ regularization to $W_{\text{key}}^{\text{s}}$, $W_{\text{val}}^{\text{s}}$, and $W_{\text{key}}^{\text{q}}$. The $L_2$-norm of these weights was scaled by 0.001 before adding it to the loss. The embedding dimension $d$ was 80. The hetero-associative memory module was represented by a square matrix of order $m = 100$ and was initialized with all its elements set to zero. Plasticity coefficients were set to $\gamma_+ = 0.01$ and $\gamma_- = 0.01$, and $w^{\text{max}}$ was set to 1. In our recurrent model, the number of memory queries $N$ was set to 3. The networks were trained for 100 epochs with a batch size of 128 (200 epochs with a batch size of 32 in the 1k training example setting). Gradients with an $L_2$-norm larger than 20.0 were normalized to have norm 20.0. Since the number of sentences and the number of words per sentence varied within and between tasks, a null symbol was used to pad them to a fixed size. The embedding of the null symbol was constraint to be zero. We observed rather high variance in the model's performance for some tasks. We therefore performed three independent runs with different random initializations and report the results of the model with the highest validation accuracy in these runs (similar to previous work [5], [6]).

## 1.3  Details to: Memory-dependent memorization

**Model details**  Here we present an extension to the H-Mem model where the computation of the value-vectors during storage is dependent on the memory content. More specifically, in the store-branch, we compute a vector $\boldsymbol{k}_t^{\text{s}}$ by passing the input-embedding $\boldsymbol{e}_t$ through a weight matrix $W_{\text{key}}^{\text{s}}$ of size $m \times d$ followed by a ReLU nonlinearity and a layer normalization:

$$\boldsymbol{k}_t^{\text{s}} = \text{LN}_{\gamma,\beta}(\text{ReLU}(W_{\text{key}}^{\text{s}}\boldsymbol{e}_t)), \tag{1}$$

where $\text{LN}_{\gamma,\beta}$ denotes layer normalization [7] with learnable parameters $\gamma$ and $\beta$, and where $\text{ReLU}(\boldsymbol{z}) = (\text{ReLU}_1(\boldsymbol{z}), \ldots, \text{ReLU}_m(\boldsymbol{z}))^{\intercal}$ with $\text{ReLU}_i(\boldsymbol{z}) = \max(0, z_i)$. We call $\boldsymbol{k}_t^{\text{s}}$ the key-vector. Similarly, we compute a value-vector $\hat{\boldsymbol{v}}_t^{\text{s}}$ by using another matrix $\hat{W}_{\text{val}}^{\text{s}}$ with the same size as $W_{\text{key}}^{\text{s}}$:

$$\hat{\boldsymbol{v}}_t^{\text{s}} = \text{ReLU}(\hat{W}_{\text{val}}^{\text{s}}\boldsymbol{e}_t). \tag{2}$$

The key-vector $\boldsymbol{k}_t^{\text{s}}$ at time $t$ is then used to extract the associated value-vector from memory by taking the matrix-vector product of $W^{\text{assoc},t}$ and the key-vector $\boldsymbol{k}_t^{\text{s}}$:

$$\tilde{\boldsymbol{v}}_t^{\text{s}} = W^{\text{assoc},t}\boldsymbol{k}_t^{\text{s}}. \tag{3}$$

The vector $\hat{\boldsymbol{v}}_t^{\mathrm{s}}$ and the vector $\tilde{\boldsymbol{v}}_t^{\mathrm{s}}$ are concatenated and passed through a matrix $W_{\mathrm{val}}^{\mathrm{s}}$ of size $m \times 2m$ followed a layer normalization to compute the final value-vector $\boldsymbol{v}_t^{\mathrm{s}}$:

$$\boldsymbol{v}_t^{\mathrm{s}} = \mathrm{LN}_{\gamma,\beta}(W_{\mathrm{val}}^{\mathrm{s}}(\hat{\boldsymbol{v}}_t^{\mathrm{s}\mathsf{T}}, \tilde{\boldsymbol{v}}_t^{\mathrm{s}\mathsf{T}})^{\mathsf{T}}). \tag{4}$$

We then use the same Hebbian plasticity rule (see Eq. (1) in the main text) for establishing the association between key-vector $\boldsymbol{k}_t^{\mathrm{s}}$ and value-vector $\boldsymbol{v}_t^{\mathrm{s}}$. The recall-branch and the final output layer were implemented as before (see *Methods* in the main text).

**Training details** We used the same hyper-parameters as before (see *Training detais* in Section 1.2), except we trained for 250 epochs using Adam [3] with a learning rate of 0.003, that was decayed exponentially (starting at epoch 150) with a decay rate of 0.01. Weights $\hat{W}_{\mathrm{val}}^{\mathrm{s}}$ were initialized using the He uniform variance scaling initializer [4]. We applied $L_2$ regularization to $\hat{W}_{\mathrm{val}}^{\mathrm{s}}$. The $L_2$-norm of these weights was scaled by 0.001 before adding it to the loss.

## 2  Results on 1k QA data set

Table S1: Test error rates (in %) on the 20 bAbI QA tasks for models using 1k training examples. Keys: BoW = bag-of-words representation; PE = position encoding representation; LE = learned encoding.

| Task | Baseline | | | H-Mem | | |
|---|---|---|---|---|---|---|
| | LSTM | MemN2N | EntNet | BoW | PE | LE |
| 1: Single Supporting Fact | 50.0 | 0.0 | 0.7 | 0.0 | 0.0 | 0.0 |
| 2: Two Supporting Facts | 80.0 | 8.3 | 56.4 | 65.5 | 66.1 | 66.7 |
| 3: Three Supporting Facts | 80.0 | 40.3 | 69.7 | 66.1 | 67.9 | 66.2 |
| 4: Two Arg. Relations | 39.0 | 2.8 | 1.4 | 43.6 | 0.0 | 0.0 |
| 5: Three Arg. Relations | 30.0 | 13.1 | 4.6 | 30.6 | 26.6 | 28.8 |
| 6: Yes/No Questions | 52.0 | 7.6 | 30.0 | 32.6 | 33.6 | 30.3 |
| 7: Counting | 51.0 | 17.3 | 22.3 | 19.3 | 18.1 | 17.6 |
| 8: Lists/Sets | 55.0 | 10.0 | 19.2 | 12.7 | 12.1 | 11.0 |
| 9: Simple Negation | 36.0 | 13.2 | 31.5 | 28.8 | 28.1 | 28.7 |
| 10: Indefinite Knowledge | 56.0 | 15.1 | 15.6 | 41.9 | 43.0 | 40.5 |
| 11: Basic Coreference | 38.0 | 0.9 | 8.0 | 2.5 | 3.3 | 2.6 |
| 12: Conjunction | 26.0 | 0.2 | 0.8 | 0.0 | 0.0 | 0.0 |
| 13: Compound Coref. | 6.0 | 0.4 | 9.0 | 4.0 | 2.0 | 3.8 |
| 14: Time Reasoning | 73.0 | 1.7 | 62.9 | 24.5 | 29.4 | 26.4 |
| 15: Basic Deduction | 79.0 | 0.0 | 57.8 | 18.8 | 0.0 | 0.0 |
| 16: Basic Induction | 77.0 | 1.3 | 53.2 | 54.2 | 55.2 | 57.0 |
| 17: Positional Reasoning | 49.0 | 51.0 | 46.4 | 41.1 | 43.9 | 44.5 |
| 18: Size Reasoning | 48.0 | 11.1 | 8.8 | 45.3 | 8.3 | 8.0 |
| 19: Path Finding | 92.0 | 82.8 | 90.4 | 88.3 | 90.0 | 86.8 |
| 20: Agent's Motivations | 9.0 | 0.0 | 2.6 | 0.0 | 0.0 | 0.0 |
| Mean error | 51.3 | 13.9 | 29.6 | 31.0 | 26.4 | 25.9 |
| Failed tasks (err. $> 5\%$) | 20 | 11 | 15 | 15 | 13 | 13 |

# 3  Comparison of our feed-forward and our recurrent model on QA data set

In Table S2 we compare our feed-forward model to our recurrent model. We compare the performance of these models in terms of their mean error, error on individual tasks, and the number of failed tasks. We observed a variety of tasks that could be solved by our recurrent model but not by the feed-forward model. Fig. S1 shows some examples of bAbI tasks along with the evolution of the validation error over 100 epochs of our H-Mem models on these tasks.

Table S2: Test error rates (in %) on the 20 bAbI QA tasks for our feed-forward model $N = 1$ and our recurrent model $N = 3$ using 10k training examples (mean test errors for 1k training examples are shown at the bottom). Results for $N = 3$ match those reported in Table 1 of the main manuscript. Keys: BoW = bag-of-words representation; PE = position encoding representation; LE = learned encoding.

| Task | H-Mem ($N = 1$) | | | H-Mem ($N = 3$) | | |
|---|---|---|---|---|---|---|
| | BoW | PE | LE | BoW | PE | LE |
| 1: Single Supporting Fact | 0.0 | 0.0 | 0.0 | 0.0 | 0.0 | 0.0 |
| 2: Two Supporting Facts | 63.9 | 64.9 | 64.2 | 0.2 | 0.0 | 0.2 |
| 3: Three Supporting Facts | 56.6 | 59.0 | 58.6 | 30.5 | 24.9 | 26.9 |
| 4: Two Arg. Relations | 42.5 | 0.0 | 0.0 | 37.8 | 0.0 | 0.0 |
| 5: Three Arg. Relations | 9.1 | 4.3 | 4.1 | 11.6 | 1.8 | 1.3 |
| 6: Yes/No Questions | 11.2 | 9.6 | 12.2 | 1.2 | 1.5 | 1.2 |
| 7: Counting | 0.6 | 0.6 | 0.8 | 0.5 | 6.8 | 0.8 |
| 8: Lists/Sets | 0.4 | 0.8 | 0.4 | 0.7 | 0.8 | 0.5 |
| 9: Simple Negation | 14.8 | 14.6 | 15.5 | 2.9 | 6.6 | 3.3 |
| 10: Indefinite Knowledge | 21.8 | 22.6 | 21.3 | 1.4 | 1.5 | 1.5 |
| 11: Basic Coreference | 5.4 | 1.0 | 0.1 | 0.0 | 0.0 | 0.0 |
| 12: Conjunction | 0.0 | 0.0 | 0.0 | 0.0 | 0.0 | 0.0 |
| 13: Compound Coref. | 2.3 | 3.8 | 2.3 | 0.0 | 0.0 | 0.0 |
| 14: Time Reasoning | 7.9 | 7.9 | 7.9 | 0.0 | 0.3 | 1.1 |
| 15: Basic Deduction | 14.0 | 0.4 | 1.0 | 10.6 | 0.0 | 0.0 |
| 16: Basic Induction | 53.4 | 55.3 | 54.2 | 53.6 | 54.3 | 54.8 |
| 17: Positional Reasoning | 41.2 | 38.0 | 38.8 | 38.7 | 41.1 | 28.7 |
| 18: Size Reasoning | 43.6 | 3.1 | 4.8 | 44.3 | 6.8 | 1.9 |
| 19: Path Finding | 83.0 | 76.4 | 74.7 | 74.8 | 70.0 | 77.1 |
| 20: Agent's Motivations | 0.0 | 0.0 | 0.0 | 0.0 | 0.0 | 0.0 |
| Mean error | 23.6 | 18.1 | 18.0 | 15.4 | 10.8 | 10.0 |
| Failed tasks (err. $> 5\,\%$) | 14 | 9 | 9 | 8 | 7 | 4 |
| On 1k training data | | | | | | |
| Mean error | 33.2 | 28.5 | 28.2 | 31.0 | 26.4 | 25.9 |
| Failed tasks (err. $> 5\,\%$) | 17 | 16 | 16 | 15 | 13 | 13 |

**A**

Task 1: Single Supporting Fact

```
Mary moved to the bathroom.
John went to the hallway.
Daniel went back to the hallway
Sandra moved to the garden.
Where is Daniel?
```
Answer: `hallway`

**B**

Task 6: Yes/No Questions

```
Sandra journeyed to the garden.
Sandra went back to the bedrooom.
John went to the hallway.
Daniel journeyed to the bathroom.
Is Sandra in the office?
```
Answer: `no`

**C**

Task 10: Indefinite Knowledge

```
Julie is either in the cinema or the park.
Mary is in the cinema.
Bill travelled to the cinema.
Fred is in the kitchen.
Is Julie in the park?
```
Answer: `maybe`

**D**

Task 19: Path Finding

```
The kitchen is south of the office.
The bedroom is north of the office.
The bathroom is east of the office.
The bedroom is east of the hallway.
How do you go from the office to the hallway?
```
Answer: `n,w`

Figure S1: **Sample stories from the bAbI data set and evolution of the validation error of H-Mem for this task. A**) Example story from task 1 of the bAbI data set and evolution of the validation error over 100 epochs of the feed-forward (blue) and recurrent (green) H-Mem model. Both models solved this task since it requires only one memory query to answer the question. **B**) Same as in **A**) but for task 6 of the bAbI data set. The recurrent model solved this task but not the feed-forward model. **C**) Same as in **A**) but for task 10 of the bAbI data set. The recurrent model solved this task but not the feed-forward model. **D**) Same as in **A**) but for task 19 of the bAbI data set. Both models had failed to solve this task.

# 4 Similarity analysis and visualization of key-value pairs

To understand H-Mem more deeply, we analyzed the key- and value-vectors that the model extracts from input. We therefore conducted an additional experiment on bAbI task 1 and 15. After training, we reran the model over stories of these tasks and computed the cosine similarity $S_C$ of recall keys $\boldsymbol{k}_{t,n}^{\mathrm{q}}$ (resp. recalled values $\boldsymbol{v}_{t,n}^{\mathrm{a}}$) to keys $\boldsymbol{k}_t^{\mathrm{s}}$ (and values $\boldsymbol{v}_t^{\mathrm{s}}$) of previous storing operations. The results of this analysis for one story of bAbI task 1 and 15 are summarized in Fig. S2 and Fig. S3, respectively. The model learns to extract the relevant key-value pairs from the input and stores that in memory. At a query, it learns which keys are essential in order to retrieve the informative values.

Figure S2: **Key-value pairs and cosine similarity for a story of bAbI task 1. A)** Shown are the extracted keys $\boldsymbol{k}_t^{\mathrm{s}}$ and values $\boldsymbol{v}_t^{\mathrm{s}}$ for each sentence of the story, the recall key $\boldsymbol{k}_{t,1}^{\mathrm{q}}$ and the recalled value $\boldsymbol{v}_{t,1}^{\mathrm{a}}$. **B)** Cosine similarity $S_C$ of recall keys $\boldsymbol{k}_{t,n}^{\mathrm{q}}$ to keys $\boldsymbol{k}_t^{\mathrm{s}}$ (top) and cosine similarity of recalled values $\boldsymbol{v}_{t,n}^{\mathrm{a}}$ to values $\boldsymbol{v}_t^{\mathrm{s}}$ (bottom). Results are shown for the $n$th memory query (note that we set the total number of memory queries $N$ to one). The recall key $\boldsymbol{k}_{7,1}^{\mathrm{q}}$ for the question where is sandra is most similar to the key of the sentence where sandra appeared (key $\boldsymbol{k}_3^{\mathrm{s}}$). The recalled value $\boldsymbol{v}_{7,1}^{\mathrm{a}}$ is most similar to the value of the sentence in time step 3, that is the sentence that supports the correct answer (kitchen).

Figure S3: **Key-value pairs and cosine similarity for a story of bAbI task 15. A**) Shown are the extracted keys $\boldsymbol{k}_t^{\mathrm{s}}$ and values $\boldsymbol{v}_t^{\mathrm{s}}$ for each sentence of the story, the recall keys $\boldsymbol{k}_{t,n}^{\mathrm{q}}$ and the recalled values $\boldsymbol{v}_{t,n}^{\mathrm{a}}$. **B**) Cosine similarity $S_C$ of recall keys $\boldsymbol{k}_{t,n}^{\mathrm{q}}$ to keys $\boldsymbol{k}_t^{\mathrm{s}}$ (top) and cosine similarity of recalled values $\boldsymbol{v}_{t,n}^{\mathrm{a}}$ to values $\boldsymbol{v}_t^{\mathrm{s}}$ (bottom) for the $n$th memory query. The recall key $\boldsymbol{k}_{7,1}^{\mathrm{q}}$ at the first recall for the question `what is emily afraid of`, is most similar to the key of the sentence containing `emily` (key $\boldsymbol{k}_4^{\mathrm{s}}$). The recall key at the second and third memory query, that is $\boldsymbol{k}_{7,2}^{\mathrm{q}}$ and $\boldsymbol{k}_{7,3}^{\mathrm{q}}$, respectively, has a high similarity to keys of sentences that contain `emily` and `cat`. The recalled value of the first and second query ($\boldsymbol{v}_{7,1}^{\mathrm{a}}$, $\boldsymbol{v}_{7,2}^{\mathrm{a}}$) has a high similarity to values of sentences containing `cat` (values $\boldsymbol{v}_1^{\mathrm{s}}$, $\boldsymbol{v}_2^{\mathrm{s}}$, and $\boldsymbol{v}_4^{\mathrm{s}}$). The value of the third memory query, that is $\boldsymbol{v}_{7,3}^{\mathrm{a}}$, is most similar to the value of the first sentence in the story ($\boldsymbol{v}_1^{\mathrm{s}}$; that is, the sentence that supports the correct answer `wolves`).

# 5 Results on QA data set of our extended model with memory-dependent memorization

Table S3: Test error rates (in %) on the 20 bAbI QA tasks for our extended model with memory-dependent memorization using 1k training examples (left) and 10k training examples (right). Keys: BoW = bag-of-words representation; PE = position encoding representation; LE = learned encoding.

| Task | On 1k training data | | | On 10k training data | | |
|---|---|---|---|---|---|---|
| | BoW | PE | LE | BoW | PE | LE |
| 1: Single Supporting Fact | 0.0 | 0.1 | 0.0 | 0.0 | 0.0 | 0.0 |
| 2: Two Supporting Facts | 18.5 | 29.0 | 22.3 | 0.1 | 0.0 | 0.0 |
| 3: Three Supporting Facts | 76.7 | 74.6 | 75.4 | 3.6 | 3.2 | 3.7 |
| 4: Two Arg. Relations | 37.3 | 0.0 | 0.0 | 31.2 | 0.0 | 0.0 |
| 5: Three Arg. Relations | 24.1 | 18.4 | 21.9 | 3.7 | 0.3 | 0.3 |
| 6: Yes/No Questions | 40.3 | 34.1 | 39.6 | 0.4 | 1.0 | 1.0 |
| 7: Counting | 15.5 | 12.3 | 14.7 | 0.7 | 0.0 | 0.2 |
| 8: Lists/Sets | 1.6 | 2.5 | 1.2 | 0.0 | 0.0 | 0.0 |
| 9: Simple Negation | 28.9 | 30.5 | 27.9 | 0.0 | 0.3 | 0.1 |
| 10: Indefinite Knowledge | 43.3 | 42.6 | 39.7 | 0.1 | 1.0 | 1.5 |
| 11: Basic Coreference | 1.6 | 1.3 | 1.3 | 0.0 | 0.0 | 0.0 |
| 12: Conjunction | 0.0 | 0.0 | 0.1 | 0.0 | 0.0 | 0.0 |
| 13: Compound Coref. | 0.0 | 0.2 | 0.0 | 0.0 | 0.0 | 0.0 |
| 14: Time Reasoning | 19.7 | 18.3 | 19.5 | 0.0 | 0.3 | 0.4 |
| 15: Basic Deduction | 21.3 | 0.0 | 0.0 | 9.9 | 0.0 | 0.0 |
| 16: Basic Induction | 54.9 | 56.7 | 54.1 | 0.5 | 0.0 | 0.3 |
| 17: Positional Reasoning | 41.6 | 42.2 | 44.6 | 38.8 | 38.3 | 0.0 |
| 18: Size Reasoning | 47.2 | 8.1 | 8.7 | 41.6 | 0.6 | 0.1 |
| 19: Path Finding | 87.2 | 86.0 | 85.9 | 70.9 | 7.6 | 4.7 |
| 20: Agent's Motivations | 0.0 | 0.0 | 0.1 | 0.0 | 0.0 | 0.0 |
| Mean error | 28.0 | 22.8 | 22.9 | 10.1 | 2.6 | 0.6 |
| Failed tasks (err. > 5 %) | 14 | 12 | 12 | 5 | 2 | 0 |