[Reviews · NeurIPS 2020]

Review 1

Summary and Contributions: The paper discusses a neural hetero-associative memory system that incorporates fast Hebbian learning. The method is tested on a question answering task and a new image pairing task, and it outperforms LSTMs in these tasks.

Strengths: - The model is elegant. - The writing is clear. - The demonstration with the image task illustrated the capabilities of the model very nicely. - The model does well with small training sets.

Weaknesses: - The method is only compared to prior models with long-term memory on the question answering task, and doesn't perform as well as the prior MemN2N or EntNet models on that task.

Correctness: I believe the claims and methods are correct.

Clarity: The paper is very clearly written.

Relation to Prior Work: Relationships with past work are discussed. Unfortunately I am not familiar enough with this line of work to evaluate this discussion, but light searching didn't turn up any glaring omissions.

Reproducibility: Yes

Additional Feedback: - Is it essential that the key-value matrix is Hebbian? The rest of the network is trained with backpropagation and Adam. Would it also work just as well to train the key-value matrix in the same way but with a higher rate? Is it important that it's Hebbian? - How and why do the query and storage keys differ? - This likely reflects my own ignorance more than anything else, but isn't it possible to achieve good performance on the tasks in the paper with simple non-neural methods (e.g. storing key-value pairs of embeddings directly and finding the most similar key at lookup time)? Because I'm not clear on this, the practical value isn't obvious to me (in contrast e.g. to various vision problems that can't be done well by classical methods). Of course a model with limited practical value could still be important. For example, it might shed light on physiological function. However the physiological realism of this model wasn't argued (in fact the authors disclaimed it somewhat in the second paragraph of the discussion). For readers like me, it would be helpful to explain the practicality or physiological relevance in more detail. - UPDATE AFTER REBUTTAL: The authors raised convincing points and I have raised my score.


Review 2

Summary and Contributions: In this article the authors propose a novel neural network architecture, similar to other memory-augmented networks, but with the addition of an associative memory module. The introduction of this module allows the network to perform tasks that require maintenance and association over long time periods, with greater accuracy than LSTMs and many other memory-augmented networks. This appears to be one of the first attempts to augment networks with an associative memory, and may open the field to further investigation.

Strengths: The authors introduced and succinctly described the importance of an associative module, and how it interacts with the two controllers required for the network to perform the task. Because most neural networks do not contain this explicit dichotomy of associative versus supervised modules, this is a novel contribution. The authors justify their choice to use an associative module based on previous research.

Weaknesses: While the paper is overall strong and the results are exciting to those interested in biologically inspired networks, the current framing may be too narrow to interest many others in the NeurIPS community. Perhaps the authors could describe what modifications of the associative module (including non-biological inspiration) might further improve performance, or how the task-control vs memory dichotomy could be more widely used in neural network research. Following the author's rebuttal, they seem inclined to include more details regarding the biological inspiration and other details that should increase interest of the wider audience.

Correctness: The empirical methodology appears standard, and is reported in sufficient detail to recreate the results. The resulting claims are justified by the performance of the network.

Clarity: The writing is clear and succinct.

Relation to Prior Work: The authors reference prior work throughout the paper, while also emphasizing how the associative module makes the current implementation different from those approaches. One very recent paper that the authors may want to address is Le, Tran, Venkatesh, Self-attentive Associative Memory, ArXiV 2000, where a similar controller+associative memory approach is taken.

Reproducibility: Yes

Additional Feedback: The section of the discussion addressing biological plausibility and spiking neural network applications seems disjointed from the majority of the paper. Dropping this section would allow the authors to instead discuss what modifications of the associative memory module might be of interest in future research. [Following author responses]: The authors intend to address the various concerns of the reviewers, and I stand by my original score.


Review 3

Summary and Contributions: This work proposes a new architecture of memory network based on the Hebbian theory called Hebbian Memory Network (H-Mem). On an image paring task, the model showed much better results than LSTM. On most of the bAbI tasks, the model showed comparable performance to existing models.

Strengths: 1. The idea is novel. With memory module (Wassoc), the architecture innovatively combines the Hebbian rule and backpropagation. 2. The proposed model is quite simple but performed well on two tasks, which is surprising.

Weaknesses: 1. The motivation of the model is unclear. In other words, why can this model work on the two tasks? We cannot simply say it uses Hebbian rule which agrees with biological system then it should work. A reason, or intuition, from the perspective of machine learning should be provided. I want to see explanations on both tasks in the rebuttal. 2. Some important technical details are missing. First, are the image pairs (a natural image+a digital image) presented as a whole to the store branch, or separately (i.e., natural image features presented to W_key and digital image features presented to W_val)? In the latter case, it is easy to understand why the model works on this task. But from the description of the method, the former is used, which makes me confusing. In Fig 2, three pairs of images are dipicted in the storage stage. Is this the setting for real implementation or just an illustration and in practice there are M>>3 pairs of images in the storage stage? How large is M? Second, it is said 100 epochs were trained for the first task. What's the defintion of a epoch? In other words, how many samples were in one epoch and what's the definition of one "sample"? Does one "sample" refers to a natural-digital image pair or a sequence of such pairs? Third, in the first task, during the presence of the first 3 pairs of images, is there a teacher signal for the output? Since there is a delay between sample presentations, is there teacher signals during the delay? If yes, what are the teacher signals? Finally, it is said that W^assoc is not optimized, but how is (1) implemented during the training process. Note that (1) describes an iterative process while the training process is also an iterative process. How do the two process interact? These problems prevents me giving higher score to the paper.

Correctness: Maybe

Clarity: Some important technical details are missing.

Relation to Prior Work: Yes

Reproducibility: No

Additional Feedback: All of my questions have been clarified. Thanks. But an important technical detail is still unclear to me: In the first experiment, it is said that the teaching signal is given only after the query (the last image in a sequence of images), then this signal can influence the weights W_key^q by using BP algorithm. But how does this signal influence the weights W_val^S and W_key^S?


Review 4

Summary and Contributions: The paper presents a hetero-associate memory model inspired by Hebbian synaptic plasticity and applies it to a single-shot association task and question answering task.

Strengths: The paper is well positioned using relevant benchmark problems and comparisons to applicable alternatives. The review of previous work is very strong. Throughout the discussion the authors show a strong familiarity with the relevant literature. Experimental work is carefully done and the results are convincing.

Weaknesses: I would like to see a brief conclusion section. The papers seems to just trail off. Other than that, it seems like a strong contribution.

Correctness: Looks good to me.

Clarity: Few, minor grammatical errors. One more careful proof read would be good.

Relation to Prior Work: Good discussion of both similarities and differences to previous works. This is a strong foundation that makes everything else very convincing.

Reproducibility: Yes

Additional Feedback: I originally thought that this paper should be accepted and the authors' rebuttal did not sway me from that opinion. ;)

[Author Response · NeurIPS 2020]

**Rebuttal for ID 9805.** We thank all reviewers for their thoughtful comments.

**[R1.1]** *"The method is only compared to prior models with long-term memory on the [QA] task, and doesn't perform as*
*well as the prior MemN2N or EntNet models on that task."* This is expected as these are ML models with non-biological
features. Our goal was to show that simple local Hebbian plasticity can be utilized to solve many of these tasks. We
will also discuss in the revision how the model could be extended (see below). **[R1.2]** *"Is it essential that the key-value*
*matrix is Hebbian? [...] Would it also work just as well to train the key-value matrix in the same way [with backprop]*
*but with a higher rate?"* This would not work during inference (after optimization) as there are no targets and hence no
error-gradients. It is a separate (interesting in ML context) question whether one could find surrogate loss functions for
memory. Our goal was to show that simple *local* plasticity is sufficient for many tasks. Hebbian plasticity is a natural
choice for this both from a functional and biological perspective. **[R1.3]** *"How and why do the query and storage keys*
*differ?"* See analysis in [R3.1]. **[R1.4]** *"[...] isn't it possible to achieve good performance on the tasks in the paper*
*with simple non-neural methods (e.g. storing key-value pairs of embeddings directly and finding the most similar key at*
*lookup time?"* This approach is rather close to the approach of MemN2N. Note however that it is still essential for many
tasks to (a) optimize the embedding (b) perform several memory queries, and (c) post-process retrieved items. **[R1.5]**
*"[...] it would be helpful to explain the practical or physiological relevance in more detail."* Due to space constraints, we
did only briefly discuss these questions in lines 259–270. Will be expanded in the revision. In brief: (practical): (a) The
simplicity of the model leaves a lot of room for extensions for interesting paradigms of memory-based ML. (b) Current
energy-efficient neuromorphic hardware cannot implement previous Memory-augmented NNs. On the other hand,
Hebbian plasticity is already implemented in hardware (e.g., Intel's Loihi chip). (Neuroscience): We did deliberately
not relate the model to brain anatomy, as the organization of higher-level cognitive functions is still very much unknown.
However, Hebbian plasticity is well-supported by many experimental findings. In particular the Hippocampus might
play a pivotal rule for implementing a memory module as in our model. More generally, our study provides a first link
between research on memory-NNs and biologically plausible models of cognition.

**[R2.1]** *"Perhaps the authors could describe what modifications of the associative module (including non-biological*
*inspiration) might further improve performance, or how the task-control vs memory dichotomy could be more widely*
*used in neural network research."* Indeed, we believe that several rather simple extensions should improve model
performance (e.g., solve all bAbI tasks). First, computation of the key- and value vectors during storage could be made
dependent on the memory content (with a prior read). Second, to compute the output, the network cannot directly
integrate several recalled values. This ability seems essential for example in bAbI task 19 (path finding). Using a
recurrent network at the output (instead of $W_{\text{out}}$) could solve this issue. We will discuss these and further possible
extensions in the revised version. **[R2.2]** *"[...] Le, Tran, Venkatesh, Self-attentive Associative Memory, ArXiV 2000,*
*[...]"* Will be discussed. **[R2.3]** *"The section of the discussion addressing biological plausibility and spiking neural*
*network applications seems disjointed from the majority of the paper. Dropping this section would allow the authors to*
*instead discuss what modifications of the associative memory module might be of interest in future research."* Since one
more page can be used for the revised version, this will not be necessary. See also [R1.5].

**[R3.1]** *"[...], why can the model work on the two tasks? [...] A reason, or intuition, from the perspective of machine*
*learning should be provided."* We believe that the model learns to extract the relevant key-value pairs from the input
and stores that in memory. At a query, it learns, which keys are essential in order to retrieve the informative values. We
performed additional analysis to test this idea by computing the cosine similarity of the recall key (resp. the recalled
value) to keys (and values) of previous storing operations in bAbI task 1. The similarity of keys was $0.996 \pm 0.004$
for sentences with the same person as the person in the query ($0.020 \pm 0.028$ otherwise). For values, the similarity
was $0.981 \pm 0.026$ for sentences with the same place as the answer place ($0.323 \pm 0.119$ otherwise). This indicates
that the model learned to associate persons to places. Similar results hold for the image association task. This analysis
will be extended to other tasks and discussed in the supplement. The following points will be clarified in the revised
version: **[R3.2]** *"[...], are the image pairs [...] presented as a whole to the store branch, or separately [...]?"* They are
presented as one image as indicated in Fig. 2. The model learns which part of the image is important for the key and the
value, respectively. **[R3.3]** *"How large is M?"* We always used 3 image pairs as indicated in Fig. 2. **[R3.4]** *"What's the*
*definition of an epoch? [...], how many samples are in one epoch and what's the definition of one "sample"?"* A sample
is one full sequence of image pairs (including random images) and one query image (see Fig. 2, bottom right). One
epoch consists of all samples of the training set (12500 samples). **[R3.5]** *"[...], in the first task, during the presence of*
*the first 3 pairs of images, is there a teacher signal for the output? [...], is there a teacher signal during the delay?"*
There is only a teacher signal after the query. Then the output is computed, compared to the target, and the error is
computed. **[R3.6]** *"[...], it is said that $W^{\text{assoc}}$ is not optimized, but how is (1) implemented during the training process."*
$W^{\text{assoc}}$ is a dynamic variable just like neuron activations. The matrix is updated according to Eq. (1) after the key- and
value-vector has been computed in the store-branch (see Methods). Will be discussed in the revised version.

**[R4.1]** *"I would like to see a brief conclusion section. The papers seems to just trail off. [...] Few, minor grammatical*
*errors."* We will add a conclusions paragraph to the discussion and proofread the manuscript for grammatical errors.

[Meta-Review · NeurIPS 2020]

The authors propose a novel architecture to solve memory-based question and answering task. The key idea is to use a hetero-associative memory that utilizes fast Hebbian learning. All the reviewers agree that this paper makes a strong contribution to the literature and the experimental results are sufficient to support the claims made in the paper. Based on the clarification questions sought by the reviewers, the writing could use improvement and authors have promised to address the issues raised in the additional space they will have in the final submission.